# Development and Characterization of a Molecularly Imprinted Polymer for the Selective Removal of Brilliant Green Textile Dye from River and Textile Industry Effluents

**DOI:** 10.3390/polym15183709

**Published:** 2023-09-08

**Authors:** Miguel Luna Quinto, Sabir Khan, Jaime Vega-Chacón, Bianca Mortari, Ademar Wong, Maria Del Pilar Taboada Sotomayor, Gino Picasso

**Affiliations:** 1Technology of Materials for Environmental Remediation Group (TecMARA), Faculty of Sciences, National University of Engineering, Lima 15333, Peru; mlunaquinto@gmail.com (M.L.Q.); sabirkchemist@gmail.com (S.K.); jvegac@uni.edu.pe (J.V.-C.); 2Institute of Chemistry, State University of São Paulo (UNESP), Araraquara 14800-900, SP, Brazil; bih.mortari@gmail.com (B.M.); ademar.wong@unesp.br (A.W.); m.sotomayor@unesp.br (M.D.P.T.S.); 3National Institute of Alternative Technologies for Detection, Toxicological Evaluation and Removal of Micropollutants and Radioactives (INCT-DATREM), Araraquara 14800-900, SP, Brazil; 4Department of Natural Sciences, Mathematics, and Statistics, Federal Rural University of the Semi-Arid, Mossoro 59625-900, RN, Brazil

**Keywords:** water treatment, selective recognition, triphenylmethane dye, smart polymer, kinetics, adsorption, molecularly imprinted polymer, Brilliant Green, density-functional theory

## Abstract

In this paper, we present an alternative technique for the removal of Brilliant Green dye (BG) in aqueous solutions based on the application of molecularly imprinted polymer (MIP) as a selective adsorbent for BG. The MIP was prepared by bulk radical polymerization using BG as the template; methacrylic acid (MAA) as the functional monomer, selected via computer simulations; ethylene glycol dimethacrylate (EGDMA) as cross-linker; and 2,2′-azobis(2-methylpropionitrile) (AIBN) as the radical initiator. Scanning electron microscopy (SEM) analyses of the MIP and non-molecularly imprinted polymer (NIP)—used as the control material—showed that the two polymers exhibited similar morphology in terms of shape and size; however, N_2_ sorption studies showed that the MIP displayed a much higher BET surface (three times bigger) compared to the NIP, which is clearly indicative of the adequate formation of porosity in the former. The data obtained from FTIR analysis indicated the successful formation of imprinted polymer based on the experimental procedure applied. Kinetic adsorption studies revealed that the data fitted quite well with a pseudo-second order kinetic model. The BG adsorption isotherm was effectively described by the Langmuir isotherm model. The proposed MIP exhibited high selectivity toward BG in the presence of other interfering dyes due to the presence of specific recognition sites (IF = 2.53) on its high specific surface area (112 m^2^/g). The imprinted polymer also displayed a great potential when applied for the selective removal of BG in real river water samples, with recovery ranging from 99 to 101%.

## 1. Introduction

Since prehistoric times, natural colors have been used in a wide range of activities and for diverse purposes including coloring natural fibers, painting objects, creating cave paintings, and in cosmetic products [1,2]. More recently, the widespread use of dyes—both natural and synthetic—for hair coloring, textile manufacturing, and so forth has resulted in the production of huge volumes of dye-containing wastewater, particularly in the textile industry; the lack of adequate treatment and proper disposal of this wastewater has led to a dramatic rise in environmental pollution worldwide, posing serious risks to the health of humans and other living beings as well as to the environment as a whole [3,4,5]. Synthetic dyes are classified according to the type of chromophores present in their structure [3]. Triphenylmethane dyes are characterized by a central carbon atom linked to three aromatic rings, generally accompanied by sulfonic and amino groups, which provide the dyes with good stability and a high degree of solubility in water [6]. Triphenylmethane dyes are usually applied in the textile industry for staining fabrics such as nylon, silk, and cotton, and are also employed in other industries including the food and cosmetics industries [7]. Brilliant Green (BG) is a triphenylmethane dye (Appendix A) [8] that is commonly used in a wide range of applications apart from textile dyeing; for example, this dye is used as an antiseptic in dermatological applications [9], where it has been proven to have serious side-effects [10].

Several techniques have been employed for the removal of BG in industrial effluents [7]; these techniques have included bacterial degradation [11], sonochemical degradation [12], and electrooxidation [13], among others. In addition, a panoply of adsorbing materials have also been applied for BG removal in industrial effluents; these materials include activated carbon [14,15,16], fruit waste [17], sawdust [18], sandpaper [19], nano-adsorbents [20], clay [21,22], quantum dots [23], biomass [24], and diverse composites [25,26]. Although some of these materials have been proven to be effective when applied for BG removal in industrial effluents, they have also been found to have some non-negligible limitations including complex sample preparation, long periods of analysis, requirement of large quantities of solvents, and selectivity problems.

Molecular imprinting is a biomimetic technology that is intended to mimic the antibody–antigen system which has taken millions of years to develop naturally. Molecularly imprinted polymers (MIPs) are synthetic polymers developed with molecular cavities that are complementary to an analyte in terms of size, shape, and functional groups. Molecular cavities are specific sites bound to the analyte; these cavities allow the MIP to selectively recognize the analyte. In comparison to antibodies, MIPs are relatively cheaper synthetic materials that are found to be more stable and are capable of maintaining their properties when applied under a wide range of experimental conditions including different solvents and varying pH or temperature [27]. Some of the techniques employed in the synthesis of MIPs include electropolymerization [28,29,30,31], photopolymerization [32], and heat-induced bulk polymerization, to name a few; of these techniques, heat-induced bulk polymerization appears to have gained enormous traction among researchers in the field [33,34]. Under the bulk imprinting technique, a template molecule is incorporated into the polymer matrix during polymerization; the template is then removed (typically using a solvent; for example, via Soxhlet extraction), paving the way toward the formation of the MIP; a schematic illustration of MIPs preparation is shown in Appendix A. Afterwards, the bulk polymer is crushed mechanically to form tiny particles [35,36]. Bulk polymerization is also commonly used for the development of MIP for dispersive solid-phase extraction [37,38,39].

Computational studies, often by density-functional theory (DFT) modeling, can be used to gain a better understanding of the MIP–analyte interactions, the polymerization process [28,29], and the most efficient functional monomer/template ratio [40,41,42,43,44,45], as well as to predict the best functional monomer or even the best solvent to employ [46] prior to the beginning of polymerization.

MIPs can be used in dispersive solid-phase extraction due to their small size, high surface area, good stability, and high selectivity toward the detection of target molecules [38,47,48]. Precisely, MIPs have been extensively used for the adsorption of compounds such as dyes [40,49,50], where they have exhibited high performance, especially in terms of selectivity toward the detection of specific analytes [31,51,52].

The present work reports the synthesis of an MIP for the selective adsorption of BG in an aqueous solution through radical polymerization of methacrylic acid in the presence of BG. The MIP was evaluated using a non-molecularly imprinted polymer (NIP) as a control material. Adsorption was studied kinetically. The studies also assess the selectivity of the MIP that were deliberately spiked, and recovery tests were performed using river samples and industrial effluents spiked with BG.

## 2. Experimental Section

### 2.1. Reagents and Aqueous Samples

The starting material employed included the following: methacrylic acid (MAA), 99%; ethylene glycol dimethacrylate (EGDMA), 98%; and 2,2′-azobis(2-methylpropionitrile) (AIBN), 95%; ethanol and methanol were also included. All of the reagents were purchased from Sigma-Aldrich (São Paulo, Brazil) and were used without further purification. Additionally, ultrapure water (18 MΩ cm at 25 °C) from a Milli-Q Direct-0.3 purifier (Millipore, Burlington, MA, USA) was used to perform the experiments.

### 2.2. Molecular Modeling

Molecular modeling was employed in order to help us select an adequate functional monomer so as to obtain a highly selective MIP. The molecular mechanics studies were carried out using OpenEye^®^, HyperChem^®^ 8.0.5, VIDA 3.0.0, and multiple minima hypersurfaces with the MOPAC software (http://openmopac.net/).

### 2.3. Synthesis of Molecularly Imprinted Polymer (MIP) and Non-Molecularly Imprinted Polymer (NIP)

An amount of 0.2 mmol of BG (template) and 0.8 mmol of methacrylic acid—MAA (functional monomer) were dissolved in a flask containing 10 mL of methanol. The solution was stirred for 12 h at 25 °C. Afterwards, 4.0 mmol of ethylene glycol dimethacrylate (EGDMA) and 0.06 mmol 2,2′-azobis(2-methylpropionitrile) (AIBN)—used as a cross-linker and radical initiator, respectively—were added to the solution, which was subjected to sonication for 10 min. The system was purged with nitrogen for 10 min to remove oxygen. Subsequently, the system was heated at 60 °C for 30 min under nitrogen flow. The precipitated polymer was dried, crushed, and sieved. The template removal was performed by the Soxhlet extraction technique using a solvent mixture composed of methanol and acetic acid in the ratio of 9:1, respectively. The solvent was poured in the round flask of the soxhlet extraction system and the MIP was packed in the filter paper. The solvent was heated until ebullition. The template had been considered completely removed when no signal assigned to the analyte was detected in spectrophotometric measurements of the solvent. The molecularly imprinted polymer (MIP) was dehydrated in a desiccator and stored for the conduct of further experiments. The non-molecularly imprinted polymer (NIP), used as a control material, was prepared based on the same procedure employed for the preparation of the MIP in the absence of BG.

### 2.4. Samples Characterization

The morphological analyses were performed by scanning electron microscopy (SEM). The SEM images were obtained using a JSM-7500F microscope (JEOL, Akishima, Japan) operated at 2.00 KV, with the aid of a secondary electron detector.

The specific surface areas of the polymers were calculated using nitrogen adsorption isotherms based on the application of the BET (Brunnauer–Emmett–Teller) technique. The assays were carried out using a Gemini VII Surface Area Analyzer (Micrometrics, Atlanta Georgia, GA, USA). Prior to the analysis, the samples were cleaned by degasification. To this end, dry polymer (20 mg) was placed in a sample holder in a helium atmosphere at 80 °C for 4 h.

The identification of the functional groups in all the samples was carried out by Fourier-transform infrared spectroscopy (FTIR). Dried samples were milled with KBr and compressed into a pellet. FTIR spectra were recorded using a Vertex 70 FTIR Spectrometer (Bruker, Billerica, MA, USA) operated at the spectral range of 4000–400 cm^−1^.

The Ultraviolet-visible (UV/Vis) spectroscopy measurements were performed using Cary 60 UV/Vis spectrophotometer (Agilent Technologies, Santa Clara, CA, USA). The quantification of BG aqueous solutions was performed via the application of absorbance measurements at 645 nm while at room temperature.

The surface charge of the polymers in the aqueous suspension was determined by the measurements of the zeta potential at room temperature using ZetasizerNano—ZS90 (Malvern, Tokyo, Japan). The zeta potentials were measured in the pH range of 2–11. The isoelectric point (IEP) was calculated as the pH at which the zeta potential is zero.

### 2.5. Adsorption Assays

The adsorption assays were carried out at room temperature in 2.5 mL vials, where 8.0 mg of the polymer was mixed with 2.0 mL of dye solution at pH 5. The mixture was shaken for 40 min using a rotating shaker at 300 rpm at room temperature, followed by centrifugation for 15 min. Afterwards, 1.0 mL of supernatant was collected from the mixture in order to measure the concentration of the dye by spectroscopic analysis.

### 2.6. Kinetic Studies

The adsorption capacity at time “*t*” ($Q_{t}$) was calculated based on the following equation [42]:(1)$$Q_{t}=(C_{0}-C_{t})\frac{V}{m}$$
where $C_{0}$ (mg L^−1^) is the initial concentration of BG, $C_{t\;}$(mg L^−1^) is the concentration of BG at time “*t*”, V (L) is the volume of the BG solution, and m(g) is the adsorbent mass. The kinetic adsorption curve was obtained by plotting $Q_{t}$ as a function of time. The equilibrium time was estimated from the kinetic adsorption curve as the time at which the concentration of BG did not change significantly at higher periods of time.

The kinetic data were fitted using a pseudo-second order kinetic model based on the following equation:(2)$$\frac{t}{Q_{t}}=\frac{t}{Q_{e}}+\frac{1}{k_{2}Q_{e}^{2}}$$
where $Q_{e}$ (mg g^−1^) is the adsorption capacity at equilibrium and $k_{2}\;$(g mg^−1^ min^−1^) is the pseudo-second-rate constant of adsorption, which can be calculated by plotting *t*/*Q_t_* as a function of time.

### 2.7. Isothermic Studies

Equilibrium adsorption isotherm curves were obtained by plotting the adsorption capacity at equilibrium ($Q_{e}$) as a function of BG equilibrium concentration. Langmuir and Freundlich isotherm models were used to elucidate the adsorption behavior [53,54,55,56,57].

The Langmuir isotherm model was evaluated using the following equation:(3)$$\frac{C_{e}}{Q_{e}}=\frac{C_{e}}{Q_{m}}+\frac{1}{Q_{m}k_{L}}$$

The Freundlich isotherm model was evaluated using the following equation:(4)$$\ln Q_{e}=n^{-1}\ln C_{e}+\ln k_{F}$$
where $C_{e}$ is the equilibrium concentration (mg L^−1^), $Q_{e}$ is the adsorption capacity at equilibrium, $Q_{m}$ is the monolayer adsorption capacity (mg g^−1^), $k_{L}$ is the Langmuir constant (L mg^−1^), $k_{F}$ is the Freundlich constant, and $n^{-1}$ is the surface heterogeneity factor that reveals the nature of the adsorption.

### 2.8. Influence of the pH and Polymer Mass in the Adsorption Process

An analysis was conducted in order to evaluate the effect of the pH of the dye solution on the adsorption capacity at equilibrium. The adsorption of BG was evaluated at pH 4.0, 4.5, 5.0, 5.5, and 6.0, while the mass of the polymer (8 mg), initial dye concentration (47.6 mg L^−1^), shaking time (40 min), and volume of the dye solution (2.0 mL) were kept constant.

An analysis was also performed in order to investigate the effect of the polymer mass on the retention capacity. The adsorption of BG in the polymer was evaluated via adsorption assays using the following polymer masses: 2, 4, 6, 8, 10, and 12 mg. The pH (5.0), initial dye concentration (193.1 mg L^−1^), shaking time (40 mL), and volume of the dye solution (2.0 mL) were kept constant. The retention capacity was calculated based on the following equation:(5)$$Retention=\frac{C_{o}-C_{f}}{C_{o}}\times 100\%$$
where $C_{f}$ is the BG concentration of the supernatant after the removal of the polymer by centrifugation, and *C*_0_ is the initial dye concentration.

### 2.9. Selectivity Studies

The selectivity of the proposed MIP toward the detection of BG was evaluated in the presence of common interfering dyes including Acid Violet 19 (AV19), Acid Violet 17 (AV17), Tartrazine (TZ), and Acid Red 151 (AR151). Comparative batch adsorption tests were performed using the same conditions at a dye concentration of 100 μmol L^−1^. Each dye was quantified based on its specific wavelength: BG—645 nm, AV19—545 nm, AV17—546 nm, TZ—426 nm, and AR151—517 nm. For the analysis of the selectivity of the MIP, the distribution coefficient ($K_{d}$), imprinting factor (IF), and selectivity (S) were determined using the following three equations:(6)$$K_{d}=\frac{Q_{e}}{C_{e}}$$
(7)$$IF=\frac{K_{d}(MIP)}{K_{d}(NIP)}$$
(8)$$S=\frac{IF\;(BG)}{IF\;(interferent)}$$
where $K_{d}\left(MIP\right)$ and $K_{d}(NIP)$ are the distribution coefficients of MIP and NIP, respectively, and IF (BG) and IF (interferent) represent the imprinting factor of BG and the interferent, respectively.

### 2.10. Recovery Assays

Aqueous samples collected from a local textile industry and from two local rivers, Guaçu and Batalha (located in Araraquara, Sao Paulo, Brazil), were spiked with BG in order to obtain solutions with BG concentrations of 38.6, 48.3, and 57.9 mg L^−1^. The analysis involving the adsorption of BG in the MIP and NIP was carried out based on the procedure described in Section 2.5. The recovery rate was calculated using the following equation:(9)$$Recovery=\frac{C_{D}}{C_{S}}\times 100\%$$
where $C_{D}$ and $C_{S}$ represent the concentration detected in the solution after the adsorption assay and the spiked concentration, respectively.

## 3. Results and Discussion

The computer simulations were used to estimate the in vacuo association-free energy interaction [58,59] between BG and MAA, which was found to be −148.7 ± 0.6 kJ mol^−1^. This value (which was approximately 36 kcal mol^−1^) was found to be similar to the energy value reported in previous studies on these kinds of MIP–analyte interactions [28]. Non-covalent interactions were probably present, including hydrogen bonding, because the 100 kcal mol^−1^ barrier was not found to have been crossed [60].

SEM micrographs of the MIP and NIP are depicted in Figure 1. The images show cluster-shaped submicron particles. The MIP and NIP particles were found to be fairly similar in shape and size.

Nitrogen sorption isotherms of the MIP and NIP are shown in Appendix A. Based on the analysis using the BET technique, the specific surface area obtained for the MIP was 112 m^2^ g^−1^; this was almost three times higher than the specific surface area obtained for the NIP (40 m^2^ g^−1^). These results point to a high degree of porosity in the MIP prompted by the successful removal of the template during the washing procedure [61], which led to the formation of specific sites in the imprinted polymer with the ability to adsorb BG effectively.

The results obtained from the FTIR analysis of the BG, MAA, EGDMA, MIP and NIP are shown in Figure 2. The BG spectrum exhibited bands at 1620, 1565, and 1480 cm^−1^, which corresponded to the aromatic rings in the BG [62,63]. The MAA spectrum displayed bands at 1688, 1630, and 1200 cm^−1^, which corresponded to C=O, C=C, and C-O bonds, respectively [64]. The EGDMA spectrum exhibited similar bands at 1715, 1633, and 1142 cm^−1^ [50]. Due to the similar chemical structure of the MIP and NIP, no significant differences were found between the spectra of these polymers. The absence of a band corresponding to the C=C bond (at approximately 1630 cm^−1^) in the MIP and NIP spectra pointed to the complete polymerization of MAA and EGDMA. In addition, the MIP and NIP spectra exhibited bands at 1720 and 1150 cm^−1^, which corresponded to the C=O and C-O bonds of EGDMA, respectively; this result was expected since the molar ratio of MAA: EGDMA was 1:5 [65]. The absence of the intense band of BG at 1565 cm^−1^ in the MIP spectrum indicated the absence of BG entrapped in the MIP structure.

Appendix A shows the images related to the BG adsorption tests conducted using the MIP and NIP under the same experimental conditions. It could be seen that the dye has been successfully adsorbed in the MIP; in contrast, the NIP failed to effectively adsorb the dye.

It is worth noting that the adsorption capacity of the polymers was affected to a great extent by the pH of the solution, as shown in Figure 3a. The adsorption capacity of the MIP was found to be higher than that of the NIP under the pH range evaluated; this difference is attributed to the following factors: (i) presence of cavities with specific size and shape suitable for BG adsorption in the MIP; and (ii) presence of a much larger specific surface area in the MIP compared to the NIP. The maximum adsorption capacity was observed in the pH interval 4.5–5.0; however, the adsorption capacity was found to decrease significantly at pH values higher or lower than this pH range. This behavior can be explained when one takes into account the pKa of BG and the surface charge of the MIP. Figure 3b shows the Zeta potential of the MIP as a function of pH. As can be noted in Figure 3b, the isoelectric point (IEP) obtained for the MIP was 3.5. BG presents two pKa values: 4.93 and 2.62 [66,67]. At pH values above 3.5, the surface of the MIP is negatively charged, and this promotes the adsorption of the cationic BG due to electrostatic interactions. Although an increase in the pH value leads to an increase in the negative charge of the MIP surface, the increase in pH also reduces the degree of BG ionization at pH above 4.93, causing a decrease in the electrostatic interactions between the MIP and BG and a decline in BG adsorption. In the pH interval 4.5–5.0, the MIP exhibits sufficient negative charges that favor the adsorption of cationic BG. At pH values below 4.5, one observes an increase in the degree of BG ionization and a reduction in negative charge on the MIP surface, and this impedes the adsorption of BG through electrostatic interactions [66,67]. The mechanism involving the adsorption of BG is controlled by the electrostatic interaction between BG and the polymer. Based on the results obtained from the experiments conducted, a pH of 5 was selected for the conduct of further experiments.

Adsorption assays were also used to evaluate the effect of adsorbent mass on BG adsorption using different MIP and NIP masses ranging from 2 and 12 mg. As can be observed in Figure 4, an increase in the polymer mass was found to lead to an increase in the retention capacity. In addition, an increase in the polymer mass also led to an increase in the active sites for the adsorption of the dye, which in turn enhanced the removal of the dye from the solution. This tendency was found to continue until the active sites were sufficiently capable of adsorbing all the dye (100%) [42]; this was obtained when a mass of 10 mg of polymer was employed. In contrast, the NIP was able to adsorb only 40% of the dye. The greatest difference in terms of the adsorption of BG in the MIP and NIP was observed when a mass of 8 mg of the polymers was applied; as a result, this mass was chosen to conduct the tests.

The study of molecular adsorption kinetics is essentially important for the treatment of aqueous effluents because it provides relevant information on the rate-controlling step and the mechanism of adsorption [68]. Figure 5a shows the kinetic curves of adsorption of BG on the MIP and NIP. The adsorption rate in the MIP was found to be initially high and the maximum capacity was reached at approximately 60 min. The MIP also exhibited a relatively higher binding capacity compared to the NIP due to its molecular imprinting characteristics. The adsorption of BG in the MIP fit quite well with the pseudo-second order kinetic model (plotted in the inset of Figure 5a, r^2^ = 0.9998). The values obtained for the corresponding parameters Q_e_ and k_2_ were 10.49 ± 0.05 mg g^−1^ and 0.12 ± 0.07 mg g^−1^ min^−1^, respectively.

Figure 5b shows the adsorption isotherms of the MIP and NIP. An adsorption isotherm shows the relationship between the quantity of adsorbate taken up by the adsorbent and the remaining adsorbate concentration in solution. There are different kinds of equations that can be employed for the analysis of experimental adsorption equilibrium data. The equation parameters of these equilibrium models provide relevant information regarding the adsorption mechanism, as well as the surface characteristics and affinity of the adsorbent. Both the Langmuir and Freundlich equations were applied for the analysis of BG adsorption in the MIP and NIP at 25 °C. The Langmuir model provides a better description regarding the adsorption over a monolayer [54], while the Freundlich equation is more suitable when it comes to describing non-covalent adsorption [69]. The binding constant and the adsorption capacity were estimated based on the Langmuir model by plotting C_e_Q_e_^−1^ vs. C_e_. As can be observed in the inset of Figure 5b, the plotting exercise yielded a linear plot (r^2^ = 0.9999); this implies that the experimental BG adsorption data fit rather well with the Langmuir monolayer model. Based on the plotting results, the corresponding parameters Q_m_ and k_L_ were found to be 12.2 ± 0.1 mg g^−1^ (slightly higher than the Q_e_ value obtained under the pseudo-second-order kinetic model) and 0.9 ± 0.2 L mg^−1^, respectively.

Selectivity analyses were carried out in order to determine the imprinting effect and the capacity/ability of MIP to adsorb BG exclusively. Table 1 presents the values obtained from the analysis of capacity of retention, adsorption capacity at equilibrium (Q_e_), distribution coefficient ($K_{d}$), imprinting factor (IF), and selectivity (S) for the different dyes applied in the MIP and NIP. An IF value higher than 1 in the MIP (2.53) indicates a greater affinity of the imprinted polymer for the analyte in relation to the NIP [50]. The NIP presented a pattern of behavior similar to the other dyes; this outcome evidently points to non-specific interactions. With regard to other parameters such as Q_e_ and K_d_, BG recorded much higher values compared to the other dyes; this is attributed to the presence of specific cavities for BG recognition in the molecular MIP configuration.

To evaluate the efficiency of the MIP in real samples, the material was employed as an adsorbent for the recovery of BG in river and industrial effluent samples spiked with BG. As can be observed in Table 2, the recovery rates obtained ranged between 99 and 101%; this result points to the outstanding sorption capacity of the MIP and its ability to provide specific molecular recognition sites for analyte detection. In short, the results obtained in this study essentially show that the proposed MIP-based technique is a direct, simple, and rapid analytical tool suitable for the selective detection of BG with a good degree of accuracy.

## 4. Conclusions

This work reported the successful synthesis and application of a molecularly imprinted polymer (MIP) for the selective adsorption of BG. The MIP exhibited a markedly higher specific surface area compared to the non-molecularly imprinted polymer (NIP), and this resulted in better porosity due to the formation of specific cavities in the MIP. The SEM analysis showed that the MIP was characterized by cluster-shaped submicron particles. The adsorption rate was found to be well described by a pseudo-second order kinetic model. The adsorption isotherm was well fitted with the Langmuir isotherm adsorption model; this is indicative of BG monolayer formation. The BG adsorption retention at equilibrium was found to be much higher in the MIP than in the NIP due to the higher imprinting factor in the former (2.53). In addition, the selectivity coefficients recorded for all of the interfering dyes were higher than 1.9. The tests conducted using real samples showed that the MIP was able to adsorb almost 100% of the BG in spiked river water and industrial effluent. All of these results point to the successful formation of cavities with specific sizes and shapes for BG adsorption, and clearly show that the MIP proposed in this study has an undeniably good potential for application toward the selective removal of BG in aqueous samples and for the elaboration of sensors based on MIPs.

## Figures and Tables

**Figure 1 polymers-15-03709-f001:**
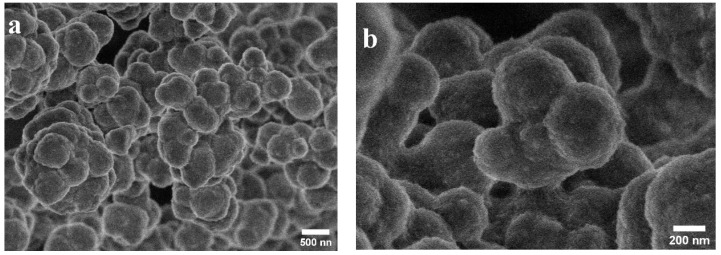
SEM Micrographs of (**a**) Molecularly Imprinted Polymer and (**b**) Non-Imprinted Polymer for Removal of Brilliant Green Textile Dye.

**Figure 2 polymers-15-03709-f002:**
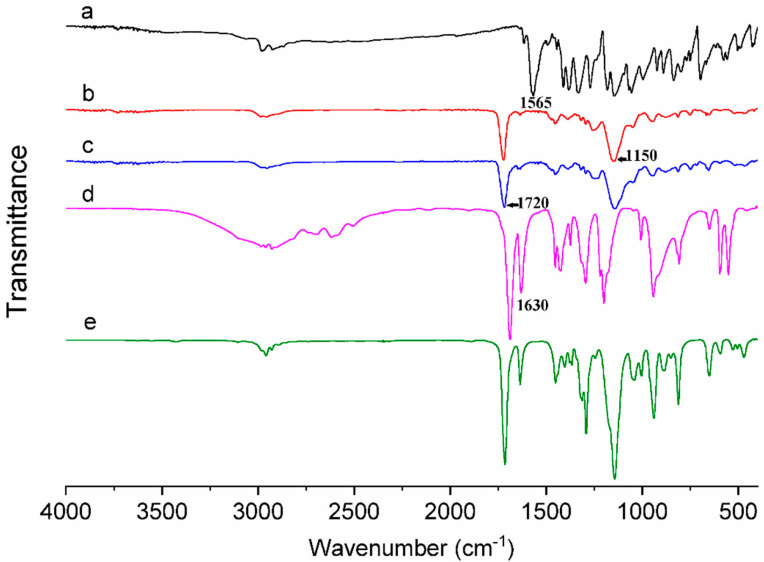
FTIR Spectra of (a) Brilliant Green Textile Dye (BG), (b) Molecularly Imprinted Polymer (MIP), (c) Non-Imprinted Polymer (NIP), (d) Methacrylic Acid (MAA), and (e) Ethylene Glycol Dimethacrylate (EGDMA).

**Figure 3 polymers-15-03709-f003:**
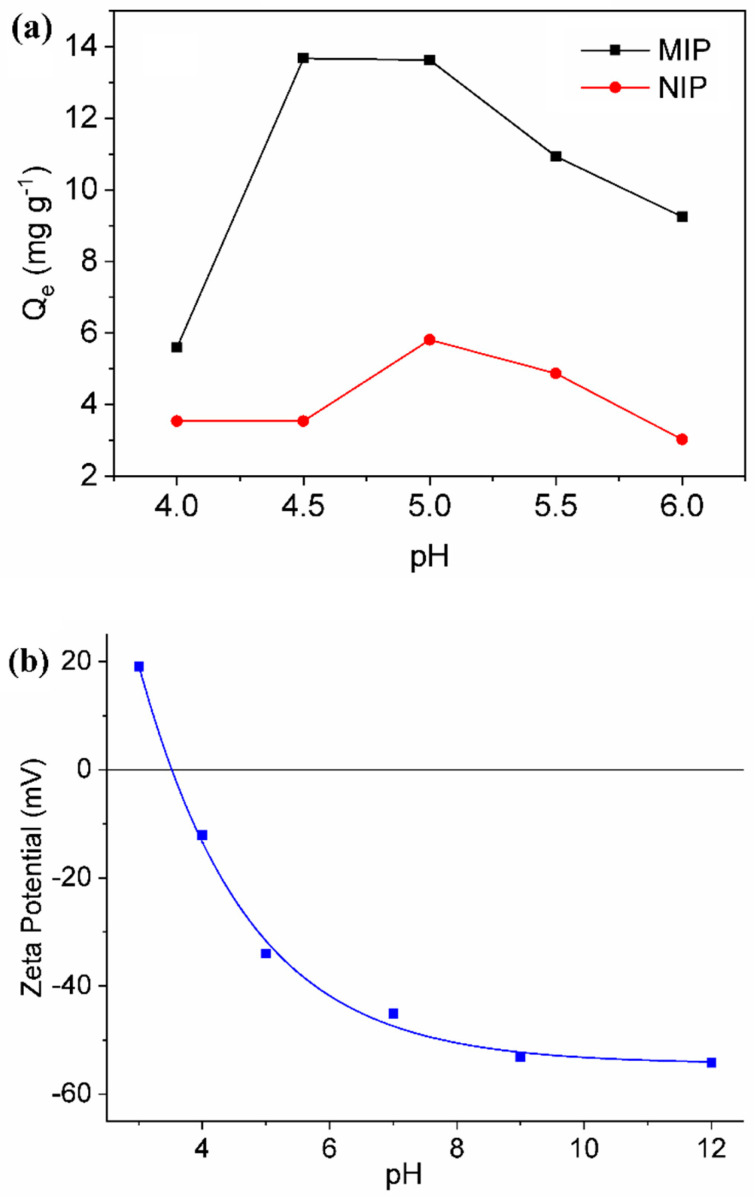
Influence of pH on the Adsorption Capacity of Molecularly Imprinted Polymer (MIP) and Non-Imprinted Polymer (NIP) for Brilliant Green Textile Dye (**a**). Zeta Potential of MIP as a Function of pH (**b**).

**Figure 4 polymers-15-03709-f004:**
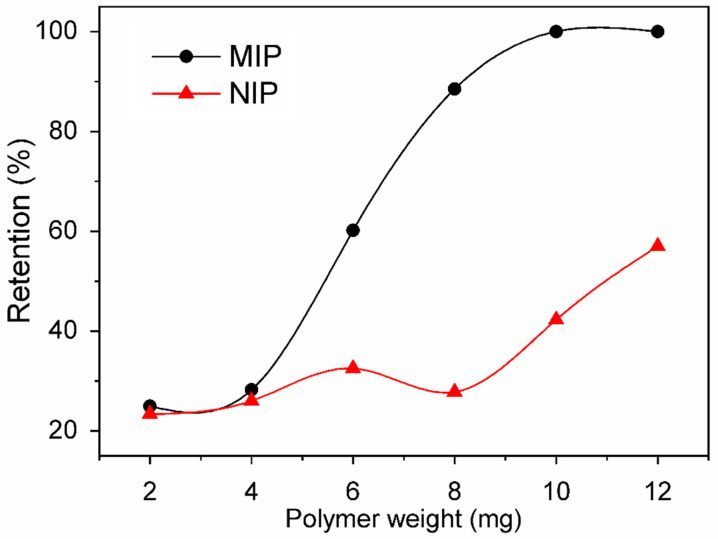
Effect of Adsorbent Mass on Retention Capacity for the Selective Adsorption of Brilliant Green Textile Dye. Experimental Conditions: Dye Solution Volume: 2.0 mL; pH: 5; BG Concentration: 193.1 mg L^−1^; Temperature: 25 °C; Time of Contact: 40 min.

**Figure 5 polymers-15-03709-f005:**
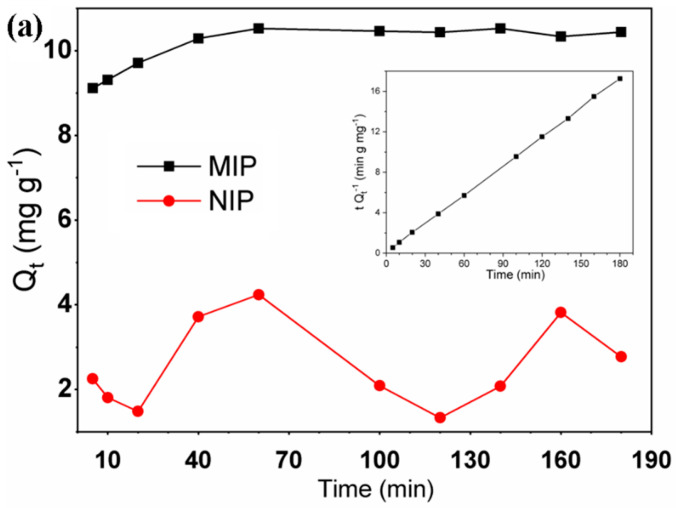
(**a**) Kinetic curves of BG adsorption on the MIP and NIP investigated. Polymer mass: 8.0 mg, dye solution volume: 2.0 mL, pH: 5, BG concentration: 48.26 mg L^−1^, and temperature: 25 °C. Inset: Pseudo-second order kinetic model applied to the MIP. (**b**) Adsorption isotherms of BG. Polymer mass: 8.0 mg, dye solution volume: 2.0 mL, pH: 5, temperature: 25 °C, and time of contact: 1 h. Inset: Equilibrium data for the MIP based on the application of the Langmuir model.

**Table 1 polymers-15-03709-t001:** Retention capacity, adsorption capacity at equilibrium (Q_e_), distribution coefficient ($K_{d}$), imprinting factor (IF), and selectivity (S) for the adsorption of different dyes applied in the MIP and NIP investigated in this study.

Dye	Retention Capacity (%)		$Q_{e}$ (mg g^−1^)		$K_{d}$ (L g^−1^)		IF	S
	MIP	NIP	MIP	NIP	MIP	NIP		
BG	87.7	34.7	10.6	4.18	0.219	0.0867	2.53	-
AV19	26.12	19.8	3.82	2.89	0.0653	0.0495	1.32	1.92
AV17	8.77	2.10	1.67	1.54	0.0219	0.0202	1.08	2.34
TZ	9.90	9.25	1.32	1.24	0.0248	0.0231	1.07	2.36
AR151	17.3	23.3	1.97	2.64	0.0433	0.0582	0.743	3.40

**Table 2 polymers-15-03709-t002:** Recovery assays. $C_{S}$ stands for spiked concentration and $C_{D}$ is the concentration detected in the solution after the adsorption assay.

Samples	$C_{S}$ (mg L^−1^)	$C_{D}$	Recovery (%)
Textile industry effluent	38.6	38.7 ± 0.1	100.2 ± 0.3
	48.3	48.1 ± 0.1	99.6 ± 0.3
	57.9	58.1 ± 0.1	100.3 ± 0.3
Guaçu river	38.6	39.0 ± 0.1	101.0 ± 0.3
	48.3	47.9 ± 0.1	99.2 ± 0.3
	57.9	57.8 ± 0.2	99.8 ± 0.6
Batalha river	38.6	38.9 ± 0.1	100.8 ± 0.3
	48.3	48.3 ± 0.1	100.0 ± 0.3
	57.9	57.9 ± 0.1	100.0 ± 0.3

## Data Availability

The raw/processed data required to reproduce these findings are available upon request to the corresponding author.

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
