# Peer review of "Development and Characterization of a Molecularly Imprinted Polymer for the Selective Removal of Brilliant Green Textile Dye from River and Textile Industry Effluents"

_polymers, 2023, doi:10.3390/polym15183709_

Round 1

Reviewer 1 Report

The MIPs were easily prepared and realized satisfactory removal of BG from real water samples. The work can contribute to MIPs preparation and water treatment. It can be published in Polymers after some revisions according to the following suggestion.

1. Line 120-121, “The template removal was performed by the Soxhlet extraction technique ..of 9:1, respectively.” How long does the extraction spend? How to make sure the complete removal of template? Please mention in this section.

2. DFT is important and interesting, and line 214-219 should schematically shown the interaction but not just mention the value.

3. For Table 1. Please mention why choose the other four analytes for selectivity?

4. The variables should use italic but the constants should not.

5. Typical peaks values should be indicated in Figure 2. And the legend should mention a-e.

6. Line 298-333, it seems confusing. Please make suitable alternation to make the clear descriptions corresponding to the related data-figure. And, the isotherms an kinetics adsorption models are suggested to schematically shown nor list related parameters in a table.

7. Can you consider the reusability of MIPs for effective removal?

8. Can you consider the production yield of MIPs? For reference, Synthesis of multi-ion imprinted polymers based on dithizone chelation for simultaneous removal of Hg2+, Cd2+, Ni2+ and Cu2+ from aqueous solutions, RSC Adv., 2016, 6, 44087-44095.

9. Since there are the SI file, please consider well the figure/table positions. For example, Figure S3 can be move to the main-text. Several other figures can be moved to SI.

10. Conclusions seeming long and not informative, should be rewritten to contain a brief both summary and outlook.

11. Please add the schematic illustration of MIPs preparation for easily understanding and citing the work.

12. Several recently related work should be cited to strengthen the research background. For example,-One-pot synthesis of magnetic molecularly imprinted microspheres by RAFT precipitation polymerization for the fast and selective removal of 17β-estradiol, RSC Adv., 2015, 5, 10611–10618.

Please carefully check the manuscript for grammar accuracy and logical clarity. For example,

-In Abstract. “innovative molecularly imprinted polymer (MIP)” should delete innovative (line 15), as well as “percentages” (line 30).

-Keywords are given according to the journals requirements being not identical to that in Title? If not, please add  Molecularly imprinted polymer , “Brilliant Green" and “density-functional theory”.

-Line 56, “among others” ?

-Line 119, “60 C”should revise the unit.

-In the manuscript from Introduction to Conclusions, the abbreviations first occurrence defined should be use subsequently. Please do not give the mixed-use.

-Line 252, Looking at the images seems colloquial.

Author Response

Reply to reviewer1: 

I provide a point-by-point response to your commnents.

Thanks for your suggestions in order the improve the original manuscript. 

Yours. Gino Picasso 

Reviewer 2 Report

In this manuscript, the authors presented a novel solution for environmental remediation. It presents a technique for the removal of Brilliant Green dye from aqueous solutions using a molecularly imprinted polymer (MIP), and compared with non-molecularly imprinted polymer (NIP). By proposing a potentially cost-effective and environmentally friendly solution for dye removal, this research could have broad implications for the field of environmental remediation. I am favorable to the publication of this manuscript. I have following suggestions:

1. Add figure legend or figure caption for a,b,c,d,e in figure 2

2. It would be a good idea to discuss the limitations of the study and suggest areas for future research in the conclusion section

Author Response

Reply to reviewer2: 

I provide a point-by-point response to your commnents.

Thanks for your suggestions in order the improve the original manuscript. 

Yours. Gino Picasso 

Reviewer 3 Report

Development and characterization of a target specific adsorbent for textile dyes such as Brilliant Green is a well thought research area. Synthesis of molecularly imprinted polymer (MIP) with methacrylic acid functional monomer, and use of these adsorbents for selective adsorbtion for BG, using non-molecularly imprinted polymer as a control is impressive. However, there are some technical issues in the execution of the research plan (discussed as major comments). Scientific language of the manuscript needs be improved with clear and detailed description of each protocol and interpretation of results from the generated figures.
Major Comments
In general, any adsorption experiment should be conducted in a vial which is inert and does not bind or adsorb the solute such as glass tubes or Erlenmeyer flasks etc. The choice of 2 mL vials or Eppendorf’s for the adsorption experiments of a dye molecule, was not the best way for such experiments. These vials are made of polypropylene, which itself is a very good adsorbent. There are tons of literature available regarding adsorption of dyes to plastic polymer-based matrix.
Specific comments
1.    Figure 1: SEM micrographs of MIP (a and b), are of same samples with same magnification. Similarly, SEM micrographs of NIP (c and d), are of same samples with same magnification. There are no need for double images, when they do not convey any additional information. One micrograph each of MIP and NIP is fine.
2.    Figure 2 shows the spectra of samples marked as a, b, c, d, e. However, in the Figure caption there is no relevance to the sample numbers. Also, the peaks corresponding to the specific band numbers, which are considered as difference between treatments, discussed in Line number 233-245, can be highlighted in the figure for clarity of interpretation of results.
3.    Figure 5 captions are numbered as 5A and 5B, although the plots in Figure 5 are numbered as (a) and (b). Please maintain the choice of either UPPER CASE or lower case in individual figures and all the figures across the manuscript.
4.    Line no 119: Please correct the temperature unit as °C. The symbol degree is missing and without that the temperature unit makes no sense.
5.    Line 121: Please provide references for the Soxhlet extraction technique used, or else describe the techniques with more details in the supplementary file.
6.    Line 138: Please replace the word “operating” with “operated”.
7.    Line 143: The use of term “nanoparticles” in this sentence is confusing and misleading. Please use same terms throughout the manuscript.
8.    Line 179: Initial dye concentration (47.6 mg L-1) was used, but again in Line 185, initial dye concentration (193.1 mg L-1) was used. Again, Figure S3 reports experiment at concentration 48.26 mg L-1).  Please clear up the confusion regarding initial dye concentration used in the adsorption experiments.
9.    Please show the data of adsorption experiments carried out at different initial dye concentrations (maybe in Supplementary file) and depending on the results found, use the optimum concentration in all other experiments. For example authors can refer to other adsorption papers.  
10.    Figure S1: As the chemical structure is not constructed by authors, please mention the source of the image in the Figure caption.

Quality of English language is okay, but can be improved to make it more clear and concise.

Author Response

Reply to reviewer3: 

I provide a point-by-point response to your commnents.

Thanks for your suggestions in order the improve the original manuscript. 

Yours. Gino Picasso 

Round 2

Reviewer 1 Report

The authors have responded the comments and revised the manuscript. It can be accepted now. 

Reviewer 3 Report

The manuscript has improved after the revision.

However, authors have not addressed the major comment (mentioned below):

"Major Comments

In general, any adsorption experiment should be conducted in a vial which is inert and does not bind or adsorb the solute such as glass tubes or Erlenmeyer flasks etc. The choice of 2 mL vials or Eppendorf’s for the adsorption experiments of a dye molecule, was not the best way for such experiments. These vials are made of polypropylene, which itself is a very good adsorbent. There are tons of literature available regarding adsorption of dyes to plastic polymer-based matrix."

Other specific comments were addressed and changes are fine.